# Effects of Incorporation of Porous Tapioca Starch on the Quality of White Salted (Udon) Noodles

**DOI:** 10.3390/foods12081662

**Published:** 2023-04-16

**Authors:** Anju Pokharel, Randhir Kumar Jaidka, N. U. Sruthi, Rewati Raman Bhattarai

**Affiliations:** 1School of Molecular Life Science, Faculty of Science and Engineering, Curtin University, Bentley, Perth, WA 6102, Australiasruthi.nedumpil@postgrad.curtin.edu.au (N.U.S.); 2Agricultural and Food Engineering Department, Indian Institute of Technology Kharagpur, Kharagpur 721302, West Bengal, India

**Keywords:** udon noodles, porous tapioca starch, enzyme treatment, ultrasound, noodle quality

## Abstract

White salted (udon) noodles are one of the major staple foods in Asian countries, particularly in Japan. Noodle manufacturers prefer the Australian noodle wheat (ANW) varieties to produce high-quality udon noodles. However, the production of this variety has reduced significantly in recent years, thus affecting the Japanese noodle market. Noodle manufacturers often add tapioca starch to compensate for the flour scarcity; however, the noodle-eating quality and texture are significantly reduced. This study, therefore, investigated the effect of the addition of porous tapioca starch on the cooking quality and texture of udon noodles. For this, tapioca starch was initially subjected to enzyme treatment, ultrasonication, and a combination of both to produce a porous starch where a combined enzyme (0.4% alpha amylase)–ultrasound treatment (20 kHz) yielded a porous starch with increased specific surface area and better absorbent properties which are ideal for udon noodle manufacturing, Later, udon noodles were prepared using three varieties of ANW, a hard Mace variety, and commercial wheat flour by incorporating the prepared porous tapioca starch at a concentration of 5% and 10% of dry ingredients. Adding this porous starch resulted in a lower cooking time with higher water absorption and desirable lower cooking loss compared to the control sample with 5% of the porous starch chosen as the optimum formulation. Increasing the level of the porous starch reduced the hardness of the noodles whilst maintaining the desired instrumental texture. Additionally, a multivariate analysis indicated a good correlation between responses’ optimum cooking time and water absorption capacity as well as turbidity and cooking loss, and a cluster analysis grouped noodle samples prepared from different varieties into the same clusters based on the porous starch added, indicating the possibility of different market strategies to improve the quality of the udon noodles produced from different wheat varieties.

## 1. Introduction

Noodles are traditional cereal-based products popular throughout the globe and highly consumed in Asian countries. The major factors leading to the popularity of Asian noodles are affordability, convenience, and long shelf life [1]. Noodles are divided into yellow alkaline noodles and white salted noodles based on the presence or absence of alkaline salts, respectively [2]. Japanese-type white salted noodles (WSN), particularly udon noodles, have significant importance as these are premium wheat products in the market [3]. The key ingredients for udon noodles are wheat flour, salt, and water [4]. Wheat flour quality plays the most significant role in the final noodle quality. Udon noodles are made from white wheat varieties with 8 to 10% protein content and low amylose content (approx. 22% of total starch) [5]. Wheat cultivars, having desired starch functionality for udon noodles, are reported to grow only in Japan and Australia. Recently, the production of such wheat varieties reduced by 43% in 2019-20, thus affecting Japanese udon production since a major share of wheat for udon noodles is bought from Australia [6]. A lack of cultivated land, poor economies of scale, and changing growth conditions have led to Japan relying on wheat imports from Australia. Western Australia is the only state that produces suitable wheat varieties for udon-making due to appropriate agronomic conditions and judicious wheat cultivar development [7]. Wheat varieties such as Australian standard white (ASW), Australian noodle wheat (ANW), and Australian premium noodle wheat (APWN) are used to produce udon noodles. These wheat varieties resemble the Japanese cultivar Norin61, which possesses a unique flavor owing to the aldehydes and ketones generated by the activity of lipoxygenase on the unsaturated fatty acids, which are ideal for the udon noodle [8]. These varieties have partially waxy starch which upon gelatinization produces a high viscographic peak and high swelling volume that are requirements for high-quality udon noodles [9].

To help maintain the high-quality udon noodle production market in Japan, it would be beneficial to understand the significance of starch quality and its modification on noodle quality. Noodle eating quality and texture are the significant aspects of high-quality udon noodles [10]. These noodles are characterized by a strand diameter of 2.0 to 3.9 mm; a bright creamy white color when cooked; high water absorption and boiling resistance; and a soft, elastic, and smooth texture (mochi-mochi) [11]. These quality attributes and the noodle-making process are influenced by the protein and starch properties of the wheat flour, such as high swelling and peak viscosity [12], low starch gelatinization temperature, amylose content [5,13] and damaged starch content, and high gluten strength [14]. The monomeric gluten proteins gliadins impart viscosity to the dough, while glutenins are accountable for the dough’s elasticity. Starches act as fillers in the proteinaceous gluten network, and a low gelatinization temperature facilitates the cooking process, thus rendering shorter cooking times during noodle-making [15]. Thus, the degrees of adhesion and interaction between the protein matrix and the starch filler determine the rheological properties of the dough and the quality of the resultant products. Since starch constitutes the largest fraction of solids in the noodle dough, the modification of starch characteristics can be used to improve the eating quality of the udon noodle.

Native wheat starch shows bimodal size distribution with large A-type and small B-type granules with a lenticular and spherical morphology, respectively [15]. Typical cereal starch granules have some small holes and internal channels existing only on the granule surface, reducing the specific surface area that provides limited space for the adsorption and encapsulation of external substances [16]. Additionally, native starch is characterized by thermal and shear instabilities and retrogradation tendencies and is reported to exhibit an unacceptable quality when used alone for noodle-making, thus limiting their practical applications [17]. This has prompted the application of modified starches with enhanced functional properties in noodle-making. Porous starch is a novel modified and denatured starch with a honeycombed granule structure and large specific surface area characterized by numerous dents or pores of a diameter of 1 µm that is widely used in the food, chemical, cosmetics, pharmaceuticals, paper, and other industries [18]. Unlike native starch, porous starch has excellent adsorption properties due to the large specific surface area that could provide higher swelling properties. Various methods have been developed to prepare porous starch, including physical, chemical, biological, and combination methods. Ultrasonication is one such method where the acoustic energy of high frequency sound results in a cavitation phenomenon that can cause the physical depolymerization of starch [19]. However, the enzymatic method is the most commonly used due to its high catalytic efficiency and substrate specificity [16,20].

Tapioca starch is known to have numerous industrial and food applications owing to its low cost and high expansion characteristics and is reported to confer an elastic texture to noodles when modified by means of physical or chemical methods [21,22]. The present research aimed to incorporate porous tapioca starch to flour from ANW varieties to modify the overall starch quality and thereby improve the udon noodle-cooking quality while maintaining the desired extensibility in noodles. For this, porous tapioca starch was prepared using enzyme catalysis, ultrasonication, and a combination of these, and the changes in the starch surface morphology were monitored to choose the best method. Then, the effect of the addition of the porous tapioca starches on the cooking quality and textural aspects of udon noodles was studied.

## 2. Materials and Methods

### 2.1. Raw Materials

Hifumiya Udon Noodle House, Perth, WA, supplied the commercial flour for standard udon noodle preparation. The ANW (Zen, Kinsei, and Ninja) and hard (Mace) wheat varieties were obtained from Intergrain Pty Ltd., Bibra Lake, WA, and were milled and passed through a sieve to attain a particle size <180 µm in a laboratory facility at the Intergrain. Tapioca starch and salt were purchased from Coles, Waterford Plaza, WA. Alpha-amylase (Sigma A6255 from porcine pancreas) was obtained from Sigma-Aldrich (Truganina, Australia).

### 2.2. Preparation of Porous Starch

Three different treatments, viz enzyme treatment, ultrasonication, and a combination of both, were used to prepare the porous tapioca starch, as discussed by Majzoobi, Hedayati, and Farahnaky (2015) [23], with slight modifications. For all the methods, the tapioca starch suspension of 20% w/w was prepared in distilled water.

#### 2.2.1. Enzymatic Treatment

The prepared starch slurry (100 mL) was subjected to two concentrations of α-amylase enzyme, viz. 0.4% and 0.6%. The mixtures were placed in a water bath of 45 °C for 24 h for starch hydrolysis and then centrifuged(Model 5810 R, Hamburg, Germany) at 3000× *g* for 20 min at room temperature. The pellet obtained was recovered and washed three times using distilled water and dried in an oven (Contherm Scientific Limited, Wellington, NZ) at 80 °C for 12 h. The dried pellet was ground, sieved to pass 100% through a 125 µm sieve, and stored vacuum-packed for further analysis.

#### 2.2.2. Ultrasonication

The prepared starch slurry (100 mL) was subjected to ultrasonication (DSA 100-GL2, Fuzhou Desen Precision Instruments Co.,Ltd, Fuzhou, China) for 40 min using two different frequency settings: high frequency (48 kHz)/low power (60 W) and low frequency (20 kHz)/high power (100 W) at a constant amplitude of 100%. The temperature of the ultrasonication bath was maintained at 40 °C. The pellet was recovered after centrifugation at 3000× *g* for 20 min at room temperature and then dried in an oven (Thermotec 200, Contherm, Lower Hutt, New Zealand) at 80 °C for 12 h. The dried pellet was ground to pass 100% through a 125 µm sieve and stored vacuum-packed for further analysis.

#### 2.2.3. Combination Treatment

For preparing the porous starch by the combined enzymatic–ultrasound treatment, the prepared starch slurry was first treated with α-amylase (0.4% and 0.6%) (Section 2.2.1), and then the hydrolyzed suspension was treated by ultrasonication as detailed in Section 2.2.2.

### 2.3. Scanning Electron Microscopy (SEM)

The surface morphology of the prepared porous starch samples was imaged under a scanning electron microscope (Vega3 VP-SEM, Tescan, Brno–Kohoutovice, Czech Republic) in high vacuum mode with a 5 kV voltage and 5–10 mm working distance (WD) after platinum coating.

### 2.4. Udon Noodle-Making and Quality Assessment

The flour from Australian wheat cultivars and the controls (Mace and commercial) were blended with different proportions of porous tapioca starch (5 and 10% of the dry ingredients) to make the noodles. The dough crumb was made by adding a calculated amount of water and salt solution to the wheat flour–porous tapioca starch blend in a mixer. The water addition was decided based on the flour moisture content of each wheat variety using mass balance equations (calculations not shown) [3]. The crumbly dough was made to a sheet in a noodle machine with sufficient folding, and the sheets were allowed to rest at 25 °C to ensure the uniform distribution of moisture and obtain a smoother and less-streaky texture. The noodle sheets were then cut into two different strand lengths, 4 cm and 20 cm, each 2.5 ± 0.02 mm wide, and kept in a moisture-proof zip-lock plastic bag in refrigerated conditions for analysis, as discussed below.

### 2.5. Cooking Quality and Texture

#### 2.5.1. Optimal Cooking Time

The optimal cooking time (OCT) was determined in duplicate according to the AACC-approved method 66.50.01 [24]. Briefly, 20 ± 0.05 g of freshly prepared noodles (4 cm strand length) were cooked in 300 mL boiling deionised water in a beaker with constant stirring at 200 rpm using a magnetic stirrer. The beaker was covered with aluminum foil during cooking to ensure a consistent temperature and reduce the evaporation of water. The cooked noodles were squeezed between two glass slides every 30 s. The optimal cooking time was determined when the white core inside the cooked noodle disappeared.

#### 2.5.2. Water Absorption Capacity

Exactly 20 ± 0.05 g of freshly prepared noodles with a strand length of 20 cm were boiled in 300 mL distilled water to their OCT as described above. The cooked noodles were then drained, weighed, and the water absorption upon cooking was determined in duplicate [25]. Water absorption was calculated as in Equation (1).
(1)$$Water\;absorption=\frac{Weight\;of\;boiled\;noodles\;(g)\;-\;weight\;of\;raw\;noodles\;(g)}{weight\;of\;raw\;noodles\;(g)}*100$$

#### 2.5.3. Cooking Loss

Exactly 20 ± 0.05 g of freshly prepared noodles of a strand length of 20 cm were boiled in 300 mL of distilled water to their OCT as described in Section 2.5.1. The cooked noodles were drained, and the water was dried in the oven at 105 ± 1 °C for 24 h. The cooking loss percentage was calculated in duplicate [24] using Equation (2).
(2)$$Cooking\;Loss\%=\frac{Weight\;of\;Solid\;residue\;after\;drying\;(g)}{Weight\;of\;raw\;noodle\;(g)}*100$$

#### 2.5.4. Texture

The three-strand cutting system was used to determine the force required to cut through the cooked and cooled noodles with a strand length of 4 cm. A texture analyzer TA.XT2i (Stable Micro System Ltd., Godalming, UK) calibrated using 5 kg of load cell with a cutting probe of 33 mm diameter was used for the test. The equipment was set to pre-test, test, and post-test speeds of 2.0, 2.0, and 1.0 mm/s, respectively. The analysis was carried out in triplicate for each cooked sample [26].

#### 2.5.5. Turbidity

The turbidity of cooked noodles with a strand length of 20 cm was analyzed as per the method described by Jeon et al. [27]. Exactly 20 ± 0.05 g of freshly prepared noodles were cooked in 300 mL distilled water to their OCT, as described in Section 2.5.1. After cooking, the water was transferred into a 500 mL measuring cylinder, and the volume was adjusted to 500 mL using distilled water. The suspension was left undisturbed for 30 min at room temperature. Then, the absorbance was measured using a spectrophotometer (Shimadzu, Canby, CA, USA) at 675 nm immediately using the clarified supernatant portion.

### 2.6. Statistical Analysis

The data were analyzed using SPSS Version 16 software using a two-way analysis of variance (ANOVA), followed by pairwise comparisons based on marginal means and adjustment with the Bonferroni method where a *p* ≤ 0.05 was considered significant. A correlation analysis, principal component analysis (PCA), and cluster analysis were performed on the complete dataset using R software (Version 2022.07.2, Posit, Boston, MA, USA) with corrplot (), prcromp (), and factoextra packages, respectively. Scripts were written in R and ‘fviz’ and ‘ggplot2′ functions were used to visualize the multivariate analyses outputs.

## 3. Results and Discussion

### 3.1. Morphology of Starch Granules

SEM images of porous tapioca starch granules prepared using the different enzyme concentration and ultrasonication settings are presented in Figure 1.

Native tapioca starch granules (Figure 1a) generally have a spherical or ellipsoid shape with truncated ends and a relatively smooth surface [28]. Figure 1b,c show the tapioca starch granules treated with 0.4% and 0.6% α-amylase, respectively. It is evident that these enzymatic treatments resulted in fracture or surface erosion of the granules, as indicated by the red circles. A higher α-amylase concentration caused more hydrolysis and resulted in an extensively eroded surface (Figure 1c). Additionally, it was also observed that the hydrolysis proceeded in a layered manner, and the surface pores opened to the granule interior and extended to the central cavity. Similar observations were made by Chen et al. [29] and Prompiputtanapon et al. [30] upon treating cassava starch with α-amylase. However, it was also remarked that α-amylase can only hydrolyze the surface of starch granules and cannot penetrate further deep into the structure because of their endohydrolysis mechanism by which the glycosidic linkages are cleaved in a random manner rather than in a stepwise manner from the non-reducing ends [30].

Figure 1d,e show the ultrasonicated starch granules at low and high frequencies, respectively. The truncated ends, being the weakest part of the granule [31], allowed the enzymes to penetrate easily, increasing the susceptibility to enzymatic action and giving more severe changes to the surface in this granule region. Unlike enzyme treatment alone, ultrasonication alone did not induce major changes in the granule structure but gave an increased roughness of the surface (Figure 1d,e). Similar results were obtained by Sujka and Jamro [32] and Monroy et al. [33] upon ultrasonication of tuber and cereal starches. Ultrasound treatment is associated with energetic vibrations, which generate waves of high pressure and velocity that strike the starch granules causing their degradation [34]. Additionally, the water molecules in the starch slurry are split due to the sonochemical effect of power ultrasound yielding OH radicals which can attack starch granules, causing enhanced polymer degradation. Water can also easily expedite the frictional or shear forces produced as a result of shock waves due to the generation of localized hot spots through bubble collapsing. This leads to high-pressure gradients in the surrounding area sufficient to degrade the starch polymers, causing grooves, notches, and fissures on the starch granule surface [19].

Figure 1f,g show the surface of the tapioca starch granules when a combined treatment involving low-frequency ultrasound with 0.6% and 0.4% α-amylase, respectively, was employed. These treatments, in particular, ultrasound with 0.4% enzyme, gave a more porous surface than the other treatments. These pores provide the granules with a larger specific surface area to increase the absorbent properties [23]. A series of complex events that occur on the starch granules as a result of the enzyme and ultrasound treatment causes a dual modification, thus resulting in an indented surface effect. Considering the need for high water absorption in udon noodle preparation, the porous tapioca starch prepared by the addition of 0.4% α-amylase and ultrasonicated at 20 kHz for 40 min was chosen for noodle preparation owing to its highly porous surface morphology.

### 3.2. Quality Attributes of Noodles

The quality attribute values obtained for the udon noodles for each variety of wheat used and the amount of porous starch added is depicted in Figure 2. Appendix A shows the complete data obtained upon quality evaluation.

The results of the two-way ANOVA (Table 1) conducted for all the response parameters indicated that the quality attributes varied significantly (*p* < 0.05) depending on the wheat variety used and the amount of porous starch added, except for WAC, where only the wheat variety was the significant factor. The interaction effects were also significant for all the responses except turbidity and texture.

Further, pairwise comparisons were made between the factors to determine the significant difference between the means among the two factors studied, viz., the wheat variety and level of porous starch added (Appendix A).

#### 3.2.1. Optimum Cooking Time (OCT)

Cooking time is important for determining noodle quality and consumer acceptability as it decides whether the noodle is undercooked or overcooked [35]. Figure 2a shows the optimal cooking time of noodles prepared with 0, 5, and 10% added porous tapioca starch prepared by the addition of 0.4% alpha-amylase and ultrasonication at 20 kHz for 40 min as detailed in Section 3.1.

The commercial udon sample containing no modified starch showed the longest cooking time of 11.1 min at all levels of starch addition. On the addition of 10% porous starch, the OCT reduced significantly (*p* < 0.05) to 10.35 min. At the same level of porous starch, the cooking time for noodles prepared from Zen and Kinsei varieties, which are the ANW varieties, was the shortest. A pairwise comparison among the wheat varieties at a specific concentration of porous starch showed no significant differences between the ANW Kinsei and Zen varieties at both 5 and 10% porous starch (Appendix A). Similar results were, however, obtained for the ANW Ninja and the hard Mace varieties, and the commercial sample was significantly different from the other varieties in all cases. Similarly, upon comparing the OCT values obtained for each level of porous starch added within the same wheat variety used, it was seen that significant differences were obtained for all levels except between 5 and 10% porous starch (Appendix A). In general, a decrease in the cooking time was observed in all varieties upon the addition of porous starch, which was desirable in noodle-making. The porous structure of the noodles due to the addition of modified starch might have provided microchannels for water to penetrate inside the noodles, increasing the water uptake during cooking [1]. Starch porosity and microchannels result in a greater surface area and accelerate the heat and mass transfer during the process, thus lowering the time required for cooking. Additionally, porous starch incorporation could also reduce the starch’s gelatinization temperature, which can improve the cooking time [36]. Moreover, ANW varieties are usually medium-grained with partial waxy starches, which, when gelatinized, result in high swelling volumes, which can also contribute towards a lesser cooking time as compared to the noodle prepared from the commercial variety [9].

#### 3.2.2. Water Absorption Capacity (WAC)

Similar to OCT, the water absorption rate of commercial noodles was higher compared to the other samples, as shown in Figure 2b. After adding 10% porous starch, the WAC further increased in the case of the commercial noodle sample. This can be correlated to the increase in the porous structure of the noodle, as explained in the previous section. The same was the trend for the hard wheat variety, Mace; however, the WAC was less than the commercial variety. Hard wheat with a higher bran content could result in a thickened matrix that can encapsulate the starch granule, causing reduced water absorption [37]. Soft wheat genotypes have characteristic B-type starch granules, which are small and relatively larger in surface area, which allow them to combine more proteins and water, which could improve water absorption [38]. However, after the addition of porous starch, a decrease in WAC was observed in noodles prepared from the soft Zen and Ninja varieties. This could be due to the pores becoming too large and eroded, causing collapse, and decreasing the water absorption capacities [39].

A two-way ANOVA was performed to evaluate the influence of both the wheat variety and porous starch addition on the WAC values obtained for the noodles, where the interaction effect was also analyzed (Table 1). It was seen that the change in porous starch concentration had an insignificant effect (*p* > 0.05) on the WAC values, whereas the wheat variety and the interaction term had a significant influence. The WAC values obtained were then compared for each variety of wheat used for a specific concentration of porous starch added (Appendix A). Similar to OCT, the WAC values of ANW Zen and Ninja varieties showed no significant differences when no porous starch was added. At both 5 and 10% concentrations of porous starch, the ANW Kinsei and the hard Mace varieties yielded WAC values with no significant differences.

#### 3.2.3. Cooking Loss

The cooking loss observed was within 5% for all the prepared noodle samples, with the native Kinsei variety showing the most negligible value (Figure 2c). However, upon the addition of porous starch, there occurred an increase in the cooking loss, with the Mace, Zen, and Kinsei varieties showing a higher value. This might be due to the amylose content in the noodles. The higher the amount of amylose content, the lower the cooking loss. Adding modified starch increased the amount of amylose, thus increasing the cooking loss. Furthermore, the cooking loss may be due to the solubilization of loosely packed gelatinized starch in the gluten matrix [40]. Moreover, with the addition of porous starch, the gluten network could become looser and diluted, exposing more starch particles to the outside that can lead to increased cooking loss [41]. Therefore, a stable protein–starch matrix is essential to maintain the internal structure and reduce the rate of cooking loss. However, the cooking loss percentage of all noodle samples is within the commercial loss, i.e., 10%.

The wheat variety, the level of porous starch addition, and their interaction significantly influenced cooking loss, as was comprehended from the two-way ANOVA results (Table 1). Thus, a pairwise comparison was performed to determine the significant differences among the means of different individual parameters. It was seen that when no porous starch was added, there was no significant difference (*p* > 0.05) in the cooking loss values obtained for noodles prepared from Mace, Ninja, and Zen varieties (Appendix A). Upon the addition of porous starch, apart from the commercial wheat variety, all the other varieties showed an insignificant influence on the cooking loss value. A pairwise comparison was also made for the values obtained for different levels of the porous starch addition, and no significant difference was observed between the 5 and 10% concentration of porous starch in all the wheat varieties studied except for the commercial one (Appendix A).

#### 3.2.4. Turbidity

The turbidity of the cooking water indicates the degree of solid loss from the noodles upon cooking, which is not an acceptable noodle quality attribute by the consumers. Water turbidity can also indirectly evaluate the structural integrity of the noodle owing to the hot-water cooking process [42]. In the present experiment, high absorbance values indicating higher turbidity were evident in noodles containing a high amount of added porous starch, irrespective of the variety (Figure 2d). This observation is in line with the cooking loss, as the higher the intensity of the cooking loss, the higher the turbidity observed [40].

Results of the two-way ANOVA show a significant influence (*p* < 0.05) of both the wheat variety and level of porous starch addition on the turbidity values; however, the interaction effect was insignificant (Table 1). Based on the estimated marginal means, it was seen that the ANW Ninja and hard Mace variety were not significantly different at both levels of the porous starch addition (Appendix A). Significant differences were only obtained for the Zen variety against commercial, Mace, and Ninja varieties at the higher level of the porous starch addition. Similarly, except for the Mace and Zen varieties, the level of the porous starch addition showed significant differences in the turbidity values for all the wheat varieties studied (Appendix A).

#### 3.2.5. Texture Analysis

The hardness of the noodles indicates the strength of the gluten network within the noodle. Figure 2e shows that the commercial noodle’s firmness was less than other ANW varieties. The firmness significantly decreased in Mace and Kinsei noodles with the addition of porous starch. With the added porous starch, the gluten network is weakened due to the breakage of disulfide bonding within the polymeric protein subunits, resulting in a decreased hardness in the obtained noodle [41]. However, for other noodles, the texture data were not consistent, and a two-way ANOVA shows an insignificant effect of the interaction effect of variety and porous starch addition (Table 1). However, hardness values were increased in the noodles prepared from commercial and Zen varieties, which could result from a reinforced gluten network [43]. Adding the porous starch thus increased the firmness value of the noodles, making them harder [44]. Additionally, the amylose content of the noodles increased with the amount of porous starch, resulting in starch retrogradation, forming a strong network structure, and making the noodle harder [45].

### 3.3. Multivariate Analysis

To understand the relationship between the studied responses and the treatment variables, different multivariate analysis tools were employed. Figure 3a shows the Pearson correlation matrix of all the noodle quality indices studied for different wheat varieties and the porous starch added. A strong positive correlation was observed between OCT and WAC and turbidity and cooking loss. The observed positive correlation between cooking time and water absorption could be explained as a result of a loosely packed gluten network, and upon the addition of porous starch providing channels for water penetration and accelerating the water uptake during cooking and with increased cooking time, a more extended hydration can be observed in the cooked noodles [46]. However, OCT was found to have a strong negative correlation with both turbidity and cooking loss, such that upon the addition of porous starch, with a decrease in optimum cooking time, an increase in cooking loss was witnessed. This contradicts the observations made by Wang et al. [47] and Liu et al. [48], where a prolonged cooking time resulted in an increased cooking loss. Although the cooking loss was well within the acceptable limits of 10%, porous starch weakened the gluten network, causing solid matter to leach into the cooking water.

Noodles prepared by the addition of 0, 5, and 10% porous starch from the commercial variety were represented by C0, C5, and C10; from Mace were represented by M0, M5, and M10, respectively; from the Zen variety were represented by Z0, Z5, and Z10, respectively; from Kinsei were represented by K0, K5, and K10, respectively; and from the Ninja variety were represented by N0, N5, and N10, respectively.

The principal component analysis (PCA) was employed to obtain a linear and unsupervised visualization of the results by reducing the original data dimension by the orthogonal transformation of the vector space. A variance bar plot and scree plot were generated to visualize the importance of principal components (PCs) created based on the amount of variation they covered (Appendix A). Two PCs were extracted from the five dimensions whose eigen values were at least 1, and both PCs combined explained at least 80% of the variance. A biplot (Figure 3b) showing the loadings of variables (vectors) and PC scores of samples (dots) was thus created where the contribution rates of the first and second principal components were 53.4% and 30.6%, respectively. It was seen that OCT, turbidity, and cooking loss strongly influenced PC1, while texture and WAC had more say in PC2. Agreeing with Pearson’s correlation analysis, a small angle between turbidity and cooking loss vectors and OCT and WAC vectors indicated a positive correlation. Vectors OCT and texture almost met at right angles, showing they were not likely to be correlated.

In contrast, cooking loss and turbidity vectors diverged from OCT, indicating a strong negative correlation. The biplot also depicts the scores of each sample studied, and their positions within the four quadrants indicate their possible relation. For instance, N5, N10, K5, K10, and M10 are all closely located to each other, meaning they have similar PC scores.

Prior to the cluster analysis for classifying the data samples into groups of similar objects, a distance or dissimilarity matrix was created (Figure 3c) using the Euclidean classical method. The states with large dissimilarities are indicated by green, whereas those fairly similar are indicated by pink. For instance, the noodles prepared from the ANW Zen variety wheat with 5% added porous starch are greatly dissimilar from those made with ANW Kinsei variety wheat with 10% added starch. Similar observations can be made for the pairwise comparison of other data samples, thus indicating possible similarities between different wheat varieties while incorporating different concentrations of porous starch.

For further understanding the similarity among different groups of data, K-means clustering was employed by partitioning the dataset into a set of two and three groups (Figure 3d,e) after determining the optimal number of clusters using Elbow, Silhouette, and Gap statistic methods as depicted in Appendix A. Upon setting k = 2, the two clusters formed were based on the concentration of porous starch added, where Kinsei, Ninja, Zen, and Mace varieties with zero porous starch added were all grouped together, whereas the commercial variety was placed in a separate cluster. When the optimal cluster numbers were chosen as three, similar clustering was observed where 10% porous starch-added samples remained a separate cluster except for the Zen variety. In comparison, the Zen and Kinsei varieties with no added starch showed similar properties to the commercial and hard Mace varieties with 5% added starch. In a word, it can be presumed that adding porous starch to different types of wheat varieties, namely soft, hard, and commercial, can alter the properties of the final product, allowing possible market interventions to produce good-quality udon noodles.

## 4. Conclusions

This study demonstrated the new knowledge on adding porous tapioca starch to wheat flour in udon noodle formulation and showed the improved noodle-cooking and texture quality apparent from a reduced optimum cooking time and increased water absorption capacity with cooking loss within acceptable limits. It was evident that wheat varieties with different inherent properties produced noodles of acceptable quality upon adding porous starch at different concentrations. Upon evaluating the quality parameters of noodles, it was concluded that a 5% addition of porous tapioca starch was the most desirable. However, the findings from this study are limited to the most popular wheat classes for udon-making. Still, it allows for exploring other classes, such as APWN or combinations of ANW/APWN, using a similar study design. The current findings could be valuable for noodle manufacturers to produce udon noodles from hard and soft wheat flour by adding starch with varied porosity and achieving desirable features. We suggest further research on the eating quality (noodle sensory properties) of udon noodles with the added porous tapioca starch, and investigations of interactions between wheat flour components and porous starch in the noodle matrix.

## Figures and Tables

**Figure 1 foods-12-01662-f001:**
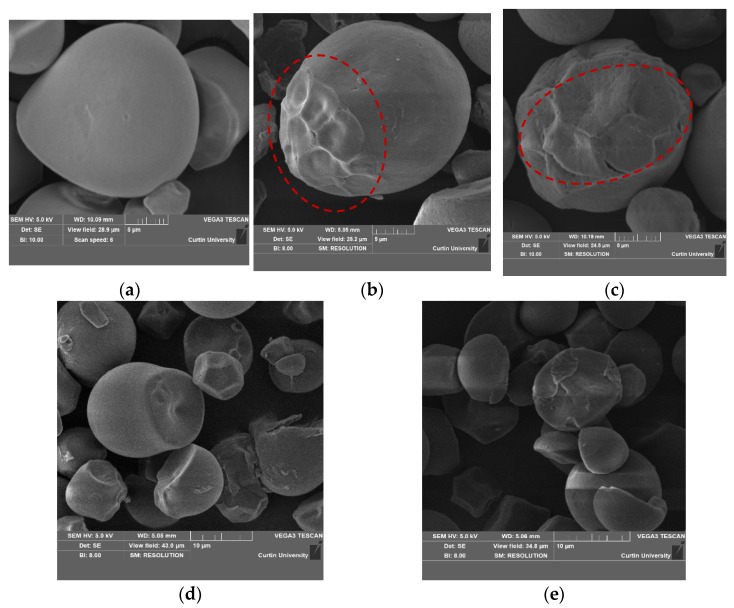
SEM images of tapioca starch granules: (**a**) native untreated, (**b**) treated with 0.4% alpha amylase, (**c**) treated with 0.6% alpha amylase, (**d**) ultrasonicated at 20 kHz, (**e**) ultrasonicated at 48 kHz, (**f**) combined 0.6% enzyme + 20 kHz ultrasonication, (**g**) combined 0.4% enzyme + 20 kHz ultrasonication. The red circle indicates the surface erosion of the granules.

**Figure 2 foods-12-01662-f002:**
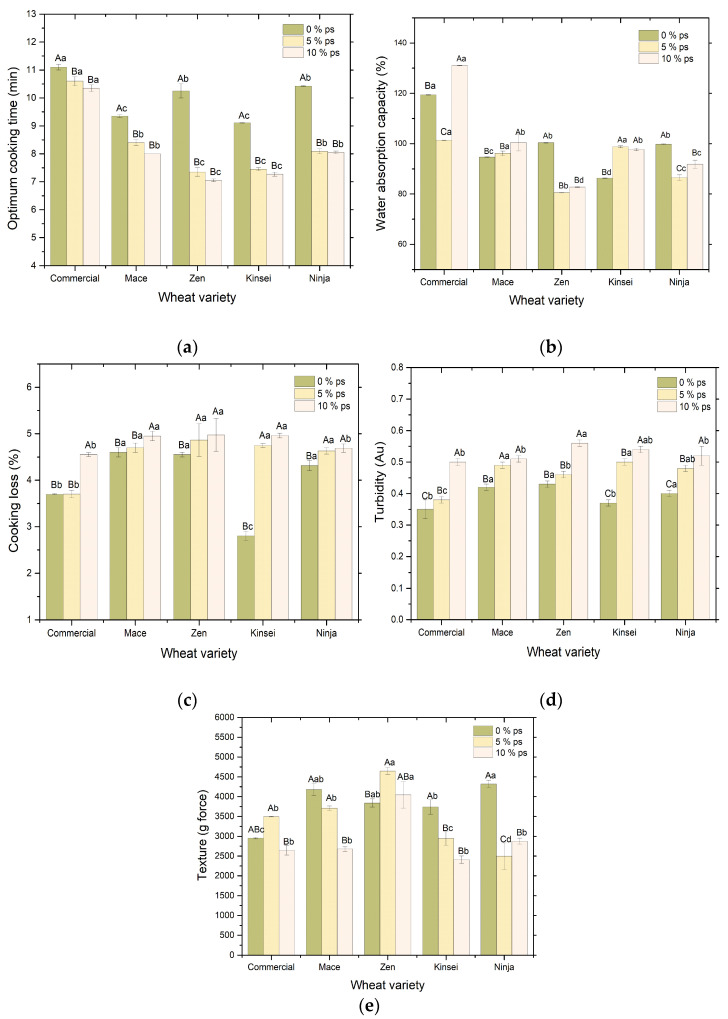
Figures showing the (**a**) optimum cooking time, (**b**) water absorption capacity, (**c**) cooking loss, (**d**) turbidity, and (**e**) texture of noodles prepared from different wheat varieties blended with 0, 5, and 10% porous starch. Different uppercase superscripts indicate significant differences in the respective quality parameters among different porous starches in the same wheat variety, while different lowercase superscripts indicate significant differences in different wheat varieties with the same level of added porous starch.

**Figure 3 foods-12-01662-f003:**
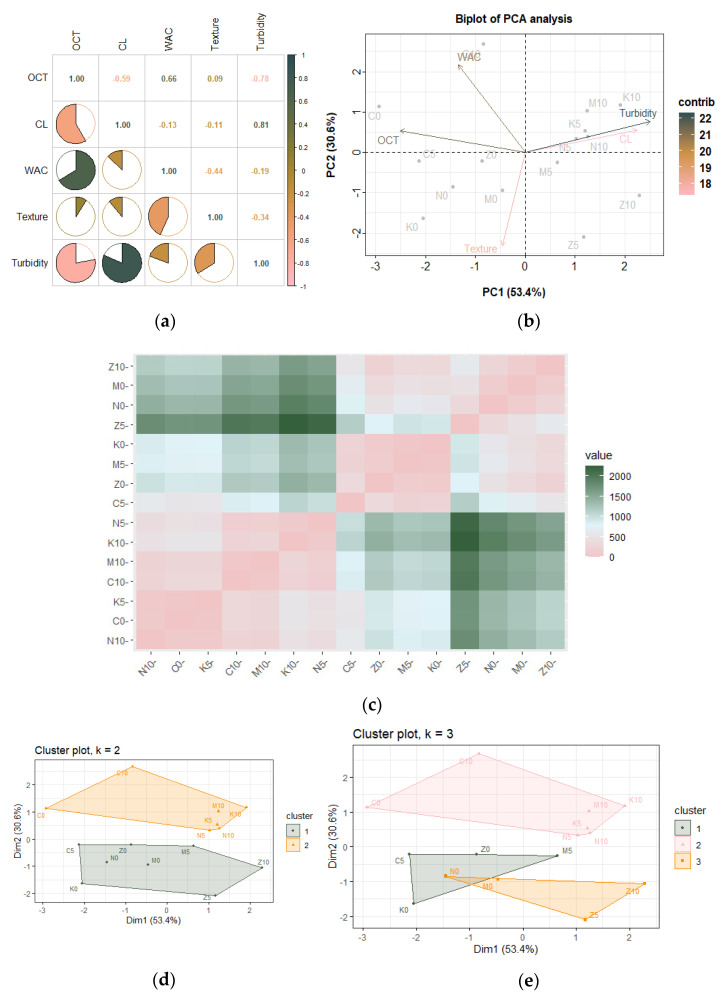
Figures showing the (**a**) Pearson correlation matrix for noodle quality indices; (**b**) PCA biplot showing the loadings of variables and individuals of each sample over the first two principal components; (**c**) pairwise distance matrices illustrating dissimilarities. K-means cluster plots with (**d**) two centers, and (**e**) three centers, at 95% confidence level.

**Table 1 foods-12-01662-t001:** Two-way ANOVA data and Fisher’s F-statistic values for different quality attributes of udon noodles with respect to the wheat variety and amount of porous starch added.

Source	OCT		WAC		Cooking Loss		Turbidity		Texture	
	MS	F	MS	F	MS	F	MS	F	MS	F
Model	5.30	56.35	417.40	10.26	0.9045	11.75	0.0114	16.39	1.178 × 10^6^	4.44
Variety	6.97	74.17	774.40	19.04	0.7785	10.11	0.0038	5.37	1.661 × 10^6^	6.26
PS	18.11	192.57	2.07	0.0510 *	3.46	44.96	0.0938	134.35	2.921 × 10^6^	11.01
Variety * PS	0.8860	9.42	168.13	4.13	0.5938	7.71	0.0014	2.00 *	5.552 × 10^5^	2.09 *
*p _model_*	<0.0001		<0.0001		<0.0001		<0.0001		0.0030	
*R^2^*	0.9690		0.8508		0.8672		0.9010		0.7116	
*Adj R^2^*	0.9519		0.7679		0.7934		0.8460		0.5514	

*: not significant at 5% level of significance; Adj: adjusted; MS: mean square. PS: porous starch added (%); OCT: optimum cooking time (min); WAC: water absorption capacity (%).

## Data Availability

All related data and methods are presented in this paper. Additional inquiries should be addressed to the corresponding author.

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
