# Peer review of "Effects of Incorporation of Porous Tapioca Starch on the Quality of White Salted (Udon) Noodles"

_foods, 2023, doi:10.3390/foods12081662_

Round 1

Reviewer 1 Report

Comments and Suggestions for Authors

The manuscript has investigated the effects of the incorporation of porous tapioca starch on the quality of white salted (Udon) noodles. The topic is interesting; However, the manuscript has several problems:

1. The abstract is unnecessarily long. Please re-write it. Also, mention the optimal formulation.

2. Please check the style of citation. For example, L 117.

3. L 145; Please express more about the noodle making.

4. According to Figure 1, the magnification of granules is not the same, which is contrary to line 143.

5. Figure 2; Please add the significant letters.

Reviewer 2 Report

Comments and Suggestions for Authors

Recommendations:

From which year came raw material?

Why not did you check the colour of noodles?

Why not did you present value of parameters in tables? In will be better.

Reviewer 3 Report

Comments and Suggestions for Authors

The work is very interesting in particular for Japanese and Australian consumers where noodles are part of the diet of their population. The contributions given in this work are not only interesting from the scientific-technological point of view but also of interest in the industrial field. For this reason, it is important, from the point of view of this reviewer, that the authors modified some aspects that are not clear in the text that they have submitted for evaluation. Basically, the authors must rewrite the abstract since the aspects related to the materials and methods are not clear, since for example, they mention that they work with 3 varieties of wheat, however in the materials and methods section 4 appear, in the same way, the characteristics of the enzymatic and ultrasonication treatments and their combination should be specified in this section. In this way, a true vision and scope can be given. Or for example, the type of tapioca used since this is not mentioned in the text.

Reviewer 4 Report

Comments and Suggestions for Authors

In this manuscript, the authors added porous tapioca starch to a wheat flour noodle formulation. The manuscript is well written and provides new insights into the potential to improve noodle quality by incorporating this type of starch. However, I have a few comments.

Firstly, in the conclusion, the authors stated that the addition of porous tapioca starch improved cooking and texture quality of the noodles. However, upon reviewing Figure 2, it is not clear which properties of the noodles were actually improved. It should be clearly stated both in the text and in the coclusion

Secondly, the process of preparing porous tapioca starch is both time and energy consuming. Therefore, it is important to consider whether the benefits of enriching the noodles with such starch are significant enough to justify this effort. It is worth noting that there are other non-cereal flours that are cheaper, healthier, and more commonly available, which could provide similar effects.

Round 2

Reviewer 4 Report

Comments and Suggestions for Authors

The authors have appropriately corrected the manuscript.